# Pin1-Catalyzed Conformation Changes Regulate Protein Ubiquitination and Degradation

**DOI:** 10.3390/cells13090731

**Published:** 2024-04-23

**Authors:** Jessica Jeong, Muhammad Usman, Yitong Li, Xiao Zhen Zhou, Kun Ping Lu

**Affiliations:** 1Departments of Biochemistry and Oncology, Schulich School of Medicine & Dentistry, Western University, London, ON N6A 5C1, Canada; jjeong63@uwo.ca (J.J.);; 2Robarts Research Institute, Western University, London, ON N6A 5B7, Canada; 3Department of Pathology and Laboratory Medicine, and Oncology, Schulich School of Medicine & Dentistry, Western University, London, ON N6A 5C1, Canada; 4Lawson Health Research Institute, Western University, London, ON N6C 2R5, Canada

**Keywords:** Pin1, phosphorylation signaling, cis–trans conformational changes, protein ubiquitination, proteosome pathway, cancer, neurodegeneration, TRIM21, E3 ligase

## Abstract

The unique prolyl isomerase Pin1 binds to and catalyzes cis–trans conformational changes of specific Ser/Thr-Pro motifs after phosphorylation, thereby playing a pivotal role in regulating the structure and function of its protein substrates. In particular, Pin1 activity regulates the affinity of a substrate for E3 ubiquitin ligases, thereby modulating the turnover of a subset of proteins and coordinating their activities after phosphorylation in both physiological and disease states. In this review, we highlight recent advancements in Pin1-regulated ubiquitination in the context of cancer and neurodegenerative disease. Specifically, Pin1 promotes cancer progression by increasing the stabilities of numerous oncoproteins and decreasing the stabilities of many tumor suppressors. Meanwhile, Pin1 plays a critical role in different neurodegenerative disorders via the regulation of protein turnover. Finally, we propose a novel therapeutic approach wherein the ubiquitin–proteasome system can be leveraged for therapy by targeting pathogenic intracellular targets for TRIM21-dependent degradation using stereospecific antibodies.

## 1. Introduction

Protein phosphorylation is one of the most important and universal regulatory post-translational modifications (PTMs) in the cell, occurring as a response to intracellular and extracellular stimuli to initiate signaling cascades or alter protein–protein interactions. In eukaryotes, this reversible mechanism consists of the covalent addition of a phosphate group at the serine, threonine, or tyrosine side chain of a protein and is mediated by kinase and phosphatase enzymes. Phosphorylation plays a crucial role in regulating a diverse range of cellular activities, one of which is ubiquitination. Specifically, phosphorylation serves to modulate a substrate’s receptivity to ubiquitination and influence the activity of substrate-ubiquitinating enzymes, particularly in relation to the ubiquitin–proteasome system (UPS) [1].

The UPS is the major enzymatic pathway in eukaryotes for intracellular protein degradation, consisting of the covalent tagging of proteins with ubiquitin (Ub) and the subsequent degradation by the 26S proteasome. Ubiquitination is orchestrated by at least three enzymes: E1: a Ub-activating enzyme; E2: a Ub-conjugating enzyme; and E3: a Ub ligase. Initially, E1 will use adenosine triphosphate (ATP) to activate Ub, creating a reactive Ub thioester at the protein C-terminus that is conjugated to E1. This activated Ub will undergo trans thiolation onto E2 before it is finally transferred onto the target substrate by E2 and E3 enzymatic activity [2]. Much of the specificity of this post-translational modification pathway is owed to this final step, wherein the E3 Ub ligase acts as a matchmaker to specifically recognize and bind the substrate to be ubiquitinated, as evidenced by the hundreds of E3 analogs required to fulfill conjugation specificity, compared to the single mammalian E1 and dozens of E2s [3,4]. E3 regulation of the UPS is particularly significant because the 26S proteasome cleaves proteins exclusively based on the Ub marker and not based on any specific amino acid sequence, like other proteases. This allows it to degrade a wide range of substrates with a high degree of specificity. Typically, substrates undergo post-translational modification with the addition of a Ub monomer (monoubiquitination); however, Ub itself can be a target for further ubiquitination, allowing the processive formation of a short Ub chain (oligoubiquitination) or a longer Ub chain (polyubiquitination) [5]. Alternatively, another class of ubiquitination factors, E4 enzymes, may also catalyze the elongation of the Ub chain [6].

One pivotal regulator of phosphorylation-mediated ubiquitination is Pin1 (protein interacting with NIMA (never in mitosis A)-1), a ubiquitously expressed and highly conserved eukaryotic peptidyl-prolyl isomerase (PPIase). Pin1 uniquely binds and catalyzes cis–trans conformational changes at specific phosphorylated Ser/Thr-Pro motifs [7,8,9], thereby modulating the activities, functions, and stabilities of a broad range of substrates after phosphorylation. Notable Pin1 substrates include several master regulators, such as p65/RelA nuclear factor-kappaB (NF-κB) [10], cyclin-dependent kinases (CDKs) [11], and the tumor suppressor protein p53 [12]. Accordingly, as Pin1 is a critical and dynamic signaling regulator for many substrates in essential functions, such as the cell cycle, immunity, and metabolism [13,14,15], it is widely distributed at both the cellular and tissue levels, depending on its substrates [16]. Thus, Pin1 can be found in both the nucleus and cytoplasm of many cell types depending on the presence of its substates, including stromal, parenchymal, and stem cells [17,18,19,20].

Pin1 is especially relevant in our discussion of phosphorylation-mediated ubiquitination given that a significant portion of the overall phosphorylation activity occurs on Ser/Thr-Pro residues, accounting for one-quarter of the sites identified in global phosphorylation studies, with pSer-Pro sites outnumbering those of pThr-Pro at a ratio of five to one [21]. Hence, Pin1 is a critical mediator in the crosstalk between ubiquitination and phosphorylation, acting as a unique and pivotal post-phosphorylation mechanism for a broad range of substrates by which the protein stability and UPS interactions can be sterically modulated (Table 1).

Pin1 functionality is critically dependent on the nature of its substrates and cellular pathways. Pin1 can interconvert from cis to trans conformation or vice versa depending on the identity of the substrate and relevant structural differences, as Ser/Thr-Pro motifs may have different preferences for being in a cis or trans configuration after phosphorylation, depending on the substrate [121]. The trans conformation of peptide bonds tends to be more energetically favorable and common due to the reduced steric hinderance from adjacent amino acids [122,123]. Consequently, Pin1 typically catalyzes cis–trans isomerizations in its substrates, although trans–cis substrates also exist [124,125,126]. For example, the cadherin CDH1 is one notable substrate wherein Pin1 trans–cis isomerization contributes to changes in stability [126].

## 2. Pin1 and the Ubiquitin–Proteasome System

Pin1 interactions with the UPS are multi-faceted and complex, as Pin1 may regulate multiple components in a pathway using any combination of several mechanisms (Figure 1). Specifically, Pin1-catalyzed isomerization may increase or decrease the stability of a protein through a variety of mechanisms, such as by (1) modulating a substrate’s compatibility with the E3 ligase through steric alterations, (2) negatively regulating substrate E3 ligases, and (3) modifying the length of a protein’s Ub chain. An interesting note is that the involvement of Pin1 in the UPS suggests that cis or trans conformation is a major factor in the regulation of Ub-mediated proteolysis.

Firstly, Pin1 increases the rate of cis–trans isomerization so that the resulting substrate has an altered affinity for E3 ligase ubiquitination. Isomerization may expose specific structural motifs that can be recognized by the E3 ligase, leading to the ubiquitination and subsequent degradation of the target substrate. Alternatively, the new conformation may bind poorly with the E3 ligase and consequently increase the substrate’s half-life [121]. For example, Pin1 recognizes and binds phosphorylated RARα, consequently promoting its protein degradation and turnover by the UPS [90]. In contrast, the opposite is also possible, where Pin1 conformational changes can enhance the protein stability. For instance, Pin1 binds and stabilizes the many members of the p53 family of transcription factors, including Tap63α and ΔNp63α [60]. Pin1 isomerization is thought to alter the cyclization of the region surrounding the PpxY motif of p63, preventing the binding of the WWP1 E3 ligase and ubiquitination [60]. As a result, Pin1 protects the levels of Tap63α and ΔNp63α and enables them to perform apoptotic and proliferative activities, respectively [60].

Secondly, Pin1 can negatively regulate many E3 ligases. One such example is the Fbw7 Skp1-Cullin-F-box (SCF)-type complex, a well-known tumor suppressor E3 ligase that targets various oncoproteins for degradation, such as c-Myc, cyclin E, XBP1, and Notch1/4 [17,127]. Pin1 promotes the self-ubiquitination and degradation of phosphorylated Fbw7 by disrupting its dimerization, likely through Fbw7 isomerization [23]. Thus, Pin1 indirectly promotes cancer development by eliminating the Fbw7 inhibition of oncogenic activities, such as the self-renewal of cancer stem cells through Notch signaling [17]. Similarly, Pin1 depletion sensitized in vitro cancer cells to Taxol by upregulating Fwb7 and subsequently decreasing the levels of the oncogene Mcl-1 [23]. Pin1 also interacts with phosphorylated c-Jun, an oncogenic transcription factor, to promote its Fbw7-mediated ubiquitination and consequent degradation [116].

Thirdly, Pin1 can regulate the protein fate and stability by controlling the length of the Ub chain. This can determine a substrate’s ability to be recognized by Ub-binding proteins [128], as well as dictate whether the Ub-conjugated substrate will proceed towards degradation pathways or associated non-proteolytic activities, such as DNA repair or signal transduction [129]. While short mono-/oligo-Ub chains commonly lead to non-proteolytic signaling activities and long poly-Ub chains lead to degradation, these functions of Ub can occur sequentially through the elongation of the Ub chain [129]. Specifically, substrates may initially undergo mono-/oligoubiquitination for cell signaling before being polyubiquitinated and degraded. While the Ub elongation factor E4 ligase often facilitates the lengthening of these chains [129,130,131], Pin1 also participates in polyubiquitination. For example, CtlP is a DNA damage factor that is recruited to the sites of double-stranded breaks to promote homologous recombination (HR) over other lower-fidelity repair mechanisms, like non-homologous end joining (NHEJ). However, the Pin1 isomerization of phosphorylated CtlP promotes its polyubiquitination and degradation [61]. By modulating the CtlP stability, Pin1 contributes to the phosphorylation-dependent regulation of DNA double-stranded repair mechanisms, indicating that abnormal Pin1 levels, like in cancer, can compromise the efficacy of the HR and NHEJ repair mechanisms and, consequently, the genomic integrity [61]. Another well-known example involves the Pin1 substrate p53, a master regulator of tumor suppressor genes, and the E3 ligase Mdm2. High Pin1 is associated with p53 monoubiquitination and nuclear export, while Pin1 inhibition results in the polyubiquitination of p53 by Mdm2 and subsequent degradation [132]. Thus, Pin1 and Mdm2 act in concert to regulate the p53 levels.

Of note, Pin1 may also regulate phosphorylation-dependent ubiquitination by regulating other PTMs, like phosphorylation, sumoylation, and acetylation, that can crosstalk with ubiquitination to coordinate cellular processes. For example, Pin1 promotes the degradation of SUV39H1, a histone methyltransferase, inhibiting its ability to induce histone 3 trimethylation and resulting in breast cancer tumor progression [109]. Similarly, post-translational modifications to Pin1 itself can regulate its activity, with the dysregulation of Pin1 PTMs contributing to the pathogenesis of cancer and Alzheimer’s Disease, as reviewed in detail by Chen and his colleagues [133].

Ultimately, by controlling the fate of phosphoproteins, Pin1 is a unique and crucial regulator of many human diseases and cellular events. In this review, we will highlight recent advancements in the understanding of Pin1-regulated ubiquitination and its role in cancer and neurodegenerative diseases. Then, we will propose how the UPS can be leveraged in the treatment of these Pin1-associated diseases by targeting pathogenic proteins for TRIM21-mediated degradation via stereospecific antibodies.

## 3. Pin1-Regulated Protein Ubiquitination in Cancer

Pin1 is tightly regulated in physiological settings and typically acts to coordinate cellular processes, like cell cycle progression and DNA repair. Typically, Pin1 is expressed at low levels in most normal cells, such as fibroblasts and epithelial cells, although it may be highly expressed in some cell subsets and certain tissues [134]. However, in most solid malignancies, like breast, prostate, and lung cancer [134,135,136], Pin1 has been found to be commonly overexpressed when compared to their physiological counterparts [134]. One study found that in 38 out of 60 human cancer types tested, Pin1 was overexpressed in 10% of these cases, suggesting that Pin1 overexpression is a critical event for tumorigenesis and may serve as an amplifier of oncogenic signaling [134]. While the mechanisms behind Pin overexpression are not fully understood, it has been posited that this trend is a result of the breakdown of the transcriptional and post-transcriptional mechanisms that usually keep Pin1 tightly regulated [137,138]. For instance, the deregulation of E2F during breast cancer promotes Pin1 expression [137], while the downregulation of the inhibitory phosphorylation of Pin1 leads to the accumulation of active Pin1 [138].

Indeed, with few exceptions, Pin1 promotes cancer progression and cancer stem cell expansion by increasing the stabilities of over 70 oncoproteins while decreasing the stabilities of over 35 tumor suppressors (Table 1) [139]. Consequently, the Pin1 levels in cancer are often correlated with poor clinical outcomes and act as a key prognostic marker in many cancer types [40,54,135,140,141]. Thus, targeting Pin1 has emerged as a viable clinical strategy, with one recent study finding that Pin1 inhibition in a pancreatic cancer model disrupted multiple cancer pathways and the immunosuppressive microenvironment, rendering it eradicable by immunochemotherapies [142]. In this section, we will explore recent discoveries in the Pin1 enhancement of oncogene stability and tumor suppressor degradation.

### 3.1. Pin1 Enhances Oncogenic Protein Stability

Excess Pin1 activity can divert substrates from degradation, enabling the enhanced stabilities of oncoproteins and the promotion of tumorigenesis by acting on cancer stem cells as well as stromal cells. One key hallmark of cancer cells is their ability to adapt the metabolic processes of both their internal environment and the tumor microenvironment to accommodate their increased proliferation and transformation [143]. One way in which Pin1-regulated ubiquitination contributes to the metabolic reprogramming of cancer cells is by stabilizing critical transcription factors, such as hypoxia inducible factor (HIF), as reviewed recently by Nakatsu and colleagues [144]. Under oxygen-deficient conditions, HIF, a master regulator of oxygen homeostasis, upregulates the stress response to promote tumor survival. HIF is composed of two subunits: one alpha subunit of which there are three subtypes (HIF-1α, HIF-2α, HIF-3α) and one beta subunit (HIF-1β) [145]. The most prominently studied subunits are the oxygen-dependent alpha subunits HIF-1α and HIF-2α, which are both necessary for cancer cell viability in the hypoxic tumor microenvironment [84]. In normoxic conditions, HIF-1α is regulated by prolyl hydroxylase domain-containing proteins (PHD) and the Von Hippel–Lindau tumor suppressor protein (pVHL), which each coordinate to promote the rapid ubiquitination and degradation of HIF-1α [146]. Consequently, HIF-1α has a short half-life of under 5 min in normoxia [147]. During hypoxia, however, HIF-1α is phosphorylated at Ser451E, allowing Pin1 to catalyze its conformational change [63,64,65]. The phosphorylation of HIF-1α is essential for its transcriptional activity and disrupts interaction with PHD- and pVHL-mediated protein degradation, significantly enhancing the stability of this oncoprotein [148]. Pin1 also interacts with CDK1-phophorylated pVHL, recruiting the E3 ligase WSB1 and facilitating pVHL ubiquitination and degradation [11]. Interrupting the Pin1/CDK1/pVHL axis results in decreased cancer cell proliferation, migration, invasion, and chemoresistance, and therefore it could be therapeutically valuable in treating cancers with wild-type VHL [11].

Furthermore, Pin1 interacts with the CDK1/2 kinase and Cul3-KLHL20 Ub ligase to degrade promyelocytic leukemia (PML), which is a tumor suppressor that typically prevents HIF-1α translation by suppressing mTOR after HIF-1α activation [100,148]. When PML is dephosphorylated at Ser518 by the phosphatase SCP1, Pin1 and KLHL20 are unable to mediate the ubiquitination and degradation of PML, thereby inhibiting the progression and growth of both in vitro and in vivo clear-cell renal carcinoma models [100]. Restored PML levels suppress the mTOR-HIF pathway, enhancing the effects of the mTOR inhibitor Temsirolimus and indicating the potential for combination treatments that target PML degradation and the mTOR pathway [149]. Additionally, Pin1 facilitates the sumoylation of PML by SUMO1 in glioma stem cells, which enables PML to interact with and stabilize c-Myc, thereby permitting the survival and carcinogenic potential of glioma stem cells [150]. Similarly, in HIF-2α, Pin1 directly binds to the subunit’s phosphorylated Ser790 site located in the nuclear export signal domain [84]. Phosphorylation at this site is essential for the stability and transactivation of HIF-2α and enables its activity even in normoxic conditions, enabling the progression of breast cancer cells [84].

Recently, Pin1 was found to promote tumor cell survival by stabilizing nuclear factor erythroid 2–related factor 2 (Nrf2). The transcription factor Nrf2 controls the expressions of genes that provide defense against both internal and external stressors, including xenobiotics and reactive oxygen species (ROS) [151]. While the transient activation of Nrf2 protects the cell from oncogenic insults, its accumulation in cancer cells allows for the constitutive induction of antioxidant and detoxifying pathways, promoting cancer proliferation in hypoxic environments and resistance to drug therapies [152]. Accordingly, Nrf2 is tightly regulated by a complex network of pathways, with the Keap1-Cul3-Rbx1 axis being the most prominent. Keap1 is an adapter subunit of the Cul3/Rbx1 E3 ligase and sequesters Nrf2 in the cytosol, leading to its continuous ubiquitination and degradation [153]. Upon encountering stressors like oxidative or electrophilic agents, Nrf2 dissociates from Keap1, allowing Pin1 to directly interact with its Ser215, Ser408, and Ser577 sites [154]. Consequently, Nrf2 is stabilized and localized to the nucleus, allowing it to transactivate cognate oncogenic pathways [83,154]. Interestingly, Pin1 also binds to the phosphorylated Ser104 and Thr277 sites of Keap1, potentially allowing Pin1 to compete with Keap1 for Nrf2 binding [154]. Furthermore, Pin1 may interact with other oncoproteins to promote Nrf2 activity. Pin1 interacts with c-Myc to bind to the Nrf2 promoter, enhancing the positive effect of c-Myc on the promoter activity [154]. Additionally, H-Ras, also known as transforming protein p21, is an oncogenic GTPase that regulates cell survival and differentiation and is correlated with increased Pin1-Nrf2 interaction [155]. Downstream, Pin1 upregulates the glutathione peroxidase 4 (GPX4) expression via Nrf2, promoting chemotherapy resistance [156]. Lastly, Pin1 has also been found to regulate the extracellular matrix composition and redox balance in pancreatic cancer through the NRF2/ARE pathway [157].

Pin1 has also been identified as a novel regulator and protein stabilizer in the oncogenic Hippo pathway, which controls the organ size by promoting cell proliferation, apoptosis, and cell stemness [158]. The Hippo pathway is divided into a tumor-suppressing serine–threonine kinase regulator module and an oncogenic transcriptional module, the latter in which Yes-associated protein (YAP) and PDZ-binding protein (TAZ) transcription factors participate [158]. Cancer cells specifically leverage YAP/TAZ to upregulate genes associated with increased stem cell renewal, cell proliferation, and apoptosis resistance. The amplification of the effects of other transcription factors, including AP-1, E2F, Myc, and CTGF, is also facilitated by YAP/TAZ, which functions as an integrator of several carcinogenic pathways, including the Wnt pathway [75,158]. Ortega and colleagues conducted a thorough review of this subject [158]. Pin1 promotes tumorigenesis by stabilizing YAP and TAZ, as well as by promoting their nuclear localization and transactivation [75,76], resulting in increased drug resistance and tumorigenicity in a breast cancer model [76]. While the exact mechanism is still being clarified, Pin1 interacts with YAP and TAZ in a phosphorylation-independent manner, suggesting that it regulates YAP/TAZ indirectly [76]. The downregulation of Pin1-YAP/TAZ activity using Cinobufacini injections is an effective treatment for osteosarcoma [159].

With regard to other components of the Hippo pathway, Pin1 also interacts with LATS1 and LATS2, two upstream kinases that phosphorylate and inhibit YAP and TAZ in the context of the anti-tubulin drug response [160]. The administration of anti-tubulin chemotherapies hyperactivates the CDK1-mediated phosphorylation of LATS2 and subsequent interactions with Pin1; the authors of this study purport that Pin1 catalyzes the conformational change of LATS1/2, switching its preferred substrate from YAP/TAZ to Pin1 and augmenting anti-tubulin-induced apoptosis [160]. Furthermore, Pin1 induces the ubiquitin-mediated degradation of STK3, a kinase that activates LATS1/2, resulting in increased tumorigenicity in melanoma mouse models [75]. As illustrated, Pin1 stabilizes a broad range of oncogenes, some of which include BRD4 in oncogenic gene expression [71], estrogen receptor-alpha in tumorigenic signaling [57], and tissue factors that can contribute to angiogenesis [72,73].

### 3.2. Pin1 Decreases Tumor Suppressor Protein Stability

Conversely, Pin1 may also disrupt the balance between oncoproteins and tumor suppressors by promoting the degradation of key tumor suppressors. Pin1 is a well-known regulator of many CDK substrates, including cyclin E, GRK2, tau, and TRF-1 [128], and it has recently been identified as a regulator of CDK substrates, like tripartite motif family 59 (TRIM59). TRIM59 is an E3 ligase and CDK5 substrate that mediates the CDK5 progression of glioblastoma. TRIM59 is phosphorylated at Ser308 by the CDK5-activated epidermal growth factor receptor (EGFR), which then recruits Pin1 to catalyze the cis–trans isomerization of TRIM59 [74]. This exposes the nuclear localization sequence of TRIM59 and facilitates its nuclear localization, wherein TRIM59 subsequently degrades macroH2A1, a tumor-suppressive histone variant [74], and inhibits TC45 from the dephosphorylation of STAT3 [74,161]. Both these mechanisms result in the sustained upregulation of STAT3 and increased glioblastoma tumorigenicity, and they are linked to a poor clinical prognosis in glioblastoma patients [74]. Similarly, TRIM32 is an E3 ligase and CDK2 substrate that mediates the CDK2 promotion of radio resistance in triple-negative breast cancer [85]. Radiotherapy stimulates the CDK2 phosphorylation of Ser328 and Ser339 on TRIM32, resulting in the subsequent Pin1 isomerization and nuclear localization of TRIM32 [85]. Nuclear TRIM32 inhibits TC45, leading to upregulated STAT3 and radio resistance in triple-negative breast cancer [85].

In addition to CDK substrates, two major regulators of cell cycle progression that interact closely with CDKs are the retinoblastoma protein (Rb) and anaphase-promoting complex (APC/C), both of which functionally collaborate to control the G1/S transition [162]. Both the APC/C and Rb are Pin1-regulated tumor suppressors that are largely inactivated in cancer [126]. Rb inhibits S-phase entry by sequestering the key transcriptions factors necessary for DNA replication during the S phase [163]. However, near the end of G1, Rb is hyperphosphorylated by cyclin–CDK complexes so that it releases chromatin-modifying enzymes and transcription factors that mediate the G1/S transition [163]. In addition, Pin1 catalyzes a conformation change in Rb, facilitating the release of the E2F transcription factor while also enabling the Rb to become fully phosphorylated [164,165,166].

Yet, Rb is frequently disrupted in many aggressive human cancers, such as triple-negative breast cancer, due to the constitutive expression of CDKs [167]. Thus, the APC/C becomes the sole regulator of the G1/S transition in these Rb-deficient cancers [126]. Recently, the involvement of Pin1 in regulating the APC/C^CDH1^ was uncovered. When bound to its CDH1 co-activator, the APC/C^CDH1^ E3 ligase typically blocks cell cycle entry by targeting various critical mitotic proteins for degradation [126]. Near the end of the G1 phase, CDKs become more active and phosphorylate CDH1 at Ser163 [126]. Pin1 then binds to CDH1 to catalyze a trans–cis isomerization that renders the co-activator resistant to dephosphorylation, inactivating the APC/C^CDH1^ and allowing mitotic proteins to accumulate and propel the cell into the S phase [126]. However, when CDH1 is not dephosphorylated, such as in the G0/G1 phase, the APC/C^CDH1^ is active and recognizes the D-box motif in the PPIase domain of Pin1, ubiquitinating Pin1 and other mitotic proteins for proteolysis and preventing S-phase entry [126]. Therefore, when Pin1 is overexpressed and CDK is constitutively expressed, like in cancer cells, the G1/S regulatory checkpoint breaks down as the APC/C^CDH1^ is inhibited and the cell proliferates unchecked [126].

Interestingly, the APC/C^CDH1^-mediated degradation of Pin1 explains why effective Pin1 inhibitors, like Sulfopin and ATRA-ATO, induce Pin1 degradation [68,142,168,169,170]. By blocking Pin1 enzymatic activity, the balance of dephosphorylated CDH1 to phosphorylated CDH1 may increase and promote APC/C^CDH1^ activation and the degradation of Pin1 [126]. Significantly, by combining Pin1 and CDK4 inhibitors to prevent APC/C^CDH1^ inactivation, the APC/C^CDH1^ induces the degradation of Pin1 and other mitotic proteins, forcing the cells into cell cycle arrest and triggering increased anti-tumor immune activity [126]. This combination treatment was highly effective in delaying cancer progression in a triple-negative breast cancer model, both disrupting the immunosuppressive tumor microenvironment and enhancing the anti-tumor immunity [126].

As illustrated, Pin1 regulates a substantial range of CDK substrates, including SEPT9 for cytokinesis completion [171], Smad3 for oncogenesis in CDK triple-negative breast cancer [172,173], and even p27, a CDK inhibitor that is regulated by Pin1 [42]. However, beyond CDK substrates, Pin1 can also directly regulate CDKs themselves, specifically CDK10. CDK10 directly interacts with the WW domain of Pin1 and becomes ubiquitinated and degraded [105]. Among other substrates, CDK10 inhibits the phosphorylation and activation of Raf-1, a key regulator for cell proliferation and survival [174]. The downregulation of CDK10 facilitates increased Raf-1 phosphorylation, resulting in increased Raf-1 activity and tamoxifen resistance in breast cancer cells [105].

Furthermore, Pin1 has also been found to regulate multiple components of the DNA repair pathway. Recently, Pin1 has also been found to promote the stability of FAAP20 in the Fanconi anemia pathway [79], to counteract excessive chromatin ubiquitination by promoting the sumoylation of the RNF168 E3 ligase [175], and to enhance the genoprotective activity of the BRCA1-BARD Ub ligase complex [78]. The Pin1 stabilization of BRCA1 also contributes to the maintenance of genomic integrity and checkpoint controls. Specifically, after ionizing radiation triggers the phosphorylation of BRCA1, Pin1 isomerizes BRCA1 at p-Ser1191 and prevents its ubiquitination at Lys1037, thereby promoting BRCA1 activity at nuclear DNA repair foci [80]. The inhibition of Pin1 removes BRCA1-mediated radio resistance and homologous recombination repair mechanisms, sensitizing breast cancer cells to radiation and PARP inhibitor treatments, which both induce stress and DNA damage that the cells can no longer fully alleviate [80].

Lastly, Pin1 was recently found to induce the ubiquitination and degradation of RUNX3, a tumor suppressor transcription factor that controls apoptosis, cell differentiation, and metastasis [110]. Pin1 binds to any of the four phosphorylated Ser/Thr-Pro sites on RUNX3 and downregulates its transcriptional activity and stability, thereby promoting breast cancer progression [110]. Similarly, tamoxifen resistance is conferred on and enhanced in breast cancer cells by Pin1-mediated SGK1 degradation [176].

## 4. Pin1-Regulated Ubiquitination in Neurodegenerative Disease

Compared to other cells in the body, Pin1 is highly expressed throughout the central and peripheral nervous systems [86,177]. While the role of Pin1 in regulating physiological neuronal functions was previously not well known, within the last decade, there has been mounting evidence of its integral role in promoting neuronal differentiation, axonal growth, synaptic transmission, and apoptosis [178]. In contrast, Pin1 attenuation in neurodegenerative diseases has been long well established, with Pin1 being most famously known to suppress Alzheimer’s Disease (AD) by regulating tau and amyloid precursor protein turnover and degradation [179,180]. In addition, the Pin1 modulation of proteins central to other neurodegenerative diseases, such as Parkinson’s Disease (PD) and Huntington’s Disease (HD), is an emerging area of research that should not be neglected. In this section, we will examine recent advancements in our understanding of Pin1’s role in negatively regulating the stability and ubiquitin-mediated degradation of proteins associated with neurodegenerative diseases, specifically AD, PD, and HD.

### 4.1. Pin1-Regulated Ubiquitination in Alzheimer’s Disease

AD is classically characterized by the abnormal aggregation of tau into neurofibrillary tangles and APP into amyloid-beta plaques. Firstly, physiological tau plays a crucial role in promoting microtubule stability and axonal outgrowth; however, when tau becomes hyperphosphorylated, it dissociates from microtubules and aggregates into the pathological neurofibrillary tangles that give rise to the synaptic dysfunction and neurodegenerative symptoms of AD [181]. Pin1 regulates tau by directly binding pThr231-Pro to catalyze a cis–trans conformation change, facilitating pThr231 dephosphorylation via protein phosphatase 2A and CDK5-p25 [182]. However, Pin1 is reduced in the AD brain by tangle sequestration [104,108], downregulation [87,183,184], phosphorylation at S71 by DAPK1 [185,186,187], oxidation at C113 [188,189], or even degradation induced by the environmental pollutant cobalt [190].

Indeed, the recent development of conformation-specific antibodies has offered the first opportunity to distinguish the cis from trans conformation of tau in AD pathogenesis. Specifically, cis-P-tau does not participate in microtubule assembly, is prone to aggregation, and resists dephosphorylation and Ub-mediated degradation, as opposed to trans-P-tau, which cannot self-associate into paired helical filaments and instead preferentially binds to microtubules to stabilize axonal transport [191]. This neuroprotective role of Pin1 is exemplified in gene manipulation studies, wherein Pin1 knockout induced tauopathy and increased tau stability in murine models while Pin1 overexpression suppressed tauopathic phenotypes [192]. Thus, Pin1 protects against tangle formation by inducing a cis–trans conformation change that restores the biological activity of phosphorylated tau to bind and promote microtubule assembly [86].

During AD, however, neurons are commonly exposed to oxidative stress, and, consequently, Pin1 may be aberrantly modified by oxidation, leading to a loss of function and polyubiquitination [107,179,193]. Furthermore, the loss of Pin1 in synapses during AD progression affects the ubiquitination of postsynaptic density proteins, a huge protein complex associated with the membrane of postsynaptic excitatory synapses that regulates synaptic plasticity [194,195]. This results in the polyubiquitination and degradation of the Shank3 protein, which regulates the organization of the postsynaptic density, ultimately leading to an increased vulnerability to the toxic effects of amyloid-beta and glutamate while also decreasing synaptic plasticity during AD progression [194].

Moerover, Pin1 is also a critical regulator of the amyloid precursor protein (APP), from which amyloid-beta (Aβ), the main component of AD-chatracteristic amyloid plaques, is derived. Physiologically, the APP plays an important role in brain development, memory, and synaptic plasticity, although its full role is not yet fully understood due to the complex manner in which the APP is processed by various secretases [196]. Pin1 binds the phosphorylated Thr668-Pro motif of the APP and catalyzes a cis–trans conformation change that reduces Aβ peptide secretion [197]. Mechanistically, it is hypothesized that, as opposed to trans-APP, the cis-APP conformation favors amyloid formation by β-secretase cleavage [197,198]. Moreover, another mechanism by which Pin1 protects against AD is by upregulating the UPS-mediated APP turnover by promoting the degradation of Glycogen Synthase Kinase-3β (GSK3β), a kinase that regulates the ubiquitination of several cancer or neurodegenerative-associated proteins in conjunction with Pin1 [177]. Other GSK3β substrates have been reviewed by Liou and colleagues [121]. By binding to Thr330-Pro, Pin1 inhibits the GSK3β phosphorylation of the APP, allowing the APP to subsequently be degraded [177]. Pin1 may also regulate GSK3β substrates; for instance, in a rat hippocampal cell culture model, Pin1 isomerized GSK3β-phosphorylated cis-HIF-1α to mediate the polyubiquitination and consequent degradation of trans-HIF-1α by the UPS [63]. This becomes therapeutically relevant in AD, wherein Pin1 and GSK3β may be downregulated under hypoxic or ischemic conditions, allowing cis-HIF-1α to accumulate and contribute to the increased expression of β-secretase 1 and the consequent amyloidogenic APP processing [63,199,200].

### 4.2. Pin1-Regulated Ubiquitination in Parkinson’s Disease

PD pathogenesis centers around the accumulation of alpha-synuclein (α-syn), a pre-synaptic neuronal protein that aberrantly misfolds into Lewy-body aggregates that perturb dopaminergic signaling, ultimately resulting in neuronal death and neurodegeneration [201,202]. Although α-syn plays a physiological role in neurotransmitter release, elevated levels of α-syn drive toxicity; thus, it is crucial for synuclein production and clearance to be tightly regulated and balanced [202]. Pin1 is commonly localized to Lewy bodies in PD brain tissue, indirectly interacting with α-syn to inhibit its degradation, thereby increasing the half-life and insolubility of α-syn and facilitating the formation of Lewy bodies [203]. In a PD murine model, the overexpression of Pin1 increased α-synuclein aggregation and upregulated pro-apoptotic cascades in dopaminergic neurons, contributing to PD onset and the associated neurodegeneration [204]. Treatment with a Pin1 inhibitor ameliorated PD-associated motor deficits, neurochemical depletion, and dopaminergic neuron degeneration in this experimental PD animal model [204]. Furthermore, Pin1 directly binds synphilin-1, a protein that participates in Lewy-body formation with α-syn, at the phosphorylated Ser-211-Pro and Ser-215-Pro motifs to promote its interaction with α-syn [204,205].

### 4.3. Pin1-Regulated Ubiquitination in Huntington’s Disease

HD is a neurodegenerative disease caused by mutant Huntingtin (Htt) protein with an elongated segment of glutamine residues [206]. Although Htt plays a critical physiological role in early development as well as in post-developmental activities, like axonal trafficking and anti-apoptotic signaling [207], mutant Htt tends to form intranuclear and cytoplasmic aggregates in neurons, especially those in the striatum, which results in cytotoxicity and HD-associated neurodegenerative symptoms [208,209]. Pin1 is playing an emerging role in HD pathogenesis. For instance, in an HD murine model, Pin1 was essential for mediating the downstream apoptotic effects of Htt, specifically by binding to the phosphorylated Ser46-Pro motif of p53, a master tumor suppressor protein, and stabilizing it into an activatable form [210]. This implies that mutant Htt may indirectly induce neuronal death by acting as an upstream inducer of Pin1-isomerized p53, rather than directly interact with it [210]. Additionally, p53 has been shown to upregulate mutant Htt expression, suggesting a positive-feedback loop [210]. Further studies by the same group using Hdh^Q111^ knock-in mice demonstrated that Pin1 is involved in HD pathogenesis throughout the lifespan of the animal [211]. In early life, Pin1 participates in the DNA damage response, presumably by stabilizing p53 [211]. For mid-age mice, Pin1 regulates hormone synthesis as well as Wnt/β-catenin signaling by stabilizing the substrate β-catenin to control neuronal differentiation [114,211]. To note, the mutant Htt stabilization of β-catenin has been demonstrated to also contribute to striatal neurotoxicity in a *Drosophila* HD model [212].

However, the ablation of Pin1 in this same Hdh^Q111^ model also decreased the mutant Htt aggregate load in the mouse striatum [211]. Similarly, a subsequent study by the same group found that Pin1 upregulated the mutant Htt clearance and reduced aggregation in an in vitro model through indirect interactions [115]. In particular, Pin1 indirectly promotes the degradation of both wild-type and mutant Htt through the UPS or autophagy [115].

Ultimately, the conflicting roles of Pin1 and the associated ubiquitination-mediated degradation in HD pathogenesis reveal the exciting therapeutic potential for targeting Pin1 manipulation, especially as previous work has demonstrated that, despite an initial impairment, the UPS functions normally during HD [213,214,215,216,217]. Moreover, moderate depletion of wild-type and mutant Htt to levels no lower than 50% does not introduce any major detrimental effects, as the mutant Htt retains some physiological activity [218,219,220,221,222,223]. This was evident in a recent clinical trial wherein a maximum reduction of 42% of mutant Htt did not yield any clinically relevant adverse effects [224]. Nevertheless, much investigation is required to elucidate the exact nature of these mechanisms in HD.

### 4.4. Pin1-Regulated Ubiquitination in Other Neurological Disorders

Beyond classical neurodegenerative diseases, Pin1-mediated ubiquitination has emerged as a regulator of adjacent neurological disorders, such as epilepsy, stroke, spinal cord injury, retinal disease, and even physiological aging, although its role is much less defined.

In epilepsy, early studies on the role of Pin1, as recently reviewed by Chen and colleagues [225], suggest that Pin1 regulates the stabilities of several disease-related proteins. For example, Pin1 binds and isomerizes protein kinase C, which is tenuously associated with the neurotransmitter balance and neuronal hyperexcitability, subsequently priming it for agonist-induced, ubiquitin-mediated degradation [107,226,227]. To note, Pin1 also regulates synaptic transmission by destabilizing and promoting the degradation of the postsynaptic density protein 95 (PSD-95), an integral regulator of synaptic plasticity and excitatory signaling, the dysregulation of which has been implicated in epileptic seizure generation [118,227,228]. As PSD-95 anchors NMDA receptors in the postsynaptic membrane, Pin1 isomerization induces a structural change that interferes with this interaction and results in a decrease in NMDA receptors, negatively affecting NMDA signaling and the dendritic spine morphology [229].

Pin1 also regulates the Notch pathway in ischemic stroke pathogenesis to promote neuronal death. In particular, Pin1 interaction with the Notch1 intracellular domain (NCID) potentiates γ-secretase cleavage that inhibits the Fwb7-induced polyubiquitination of NCID1, thereby preventing its ubiquitin-mediated proteolysis and enabling its enhancement of neuronal death after simulated ischemia in an in vitro model [45]. The subsequent treatment of Pin1-knockout mice with Pin1 inhibitors demonstrated reduced brain damage and improved functional outcomes after the induction of focal ischemic stroke [45]. In addition, Pin1 not only enhances Notch1 activity, but Notch1 also directly induces Pin1 transcription, forming a positive-feedback loop [39]. Notably, Pin1 also regulates other key players in ischemic stroke, such as NF-kβ, HIF-1α, and p53, although the crosstalk between these other molecules should not be understated [230]. For example, Pin1 directly binds to the phosphorylated Ser33/46-Pro of p53 and stabilizes it, allowing it to induce downstream apoptotic gene expression and the mitochondrial cell death pathways [28]. Specifically, rescued p53 subseqeuntly interacts with Notch to promote apoptotic pathways in neurons, comprising their viability and resistance to ischemic damage and ultimately promoting pathogenesis after ischemic stroke [231].

Moreover, in the context of spinal cord injury, Pin1 positively regulates the stability of several anti-apoptotic proteins that resist neuronal death after traumatic injury. Normally, Pin1 plays a physiological role in binding myeloid cell leukemia sequence-1 (Mcl-1) at phosphorylated Thr163-Pro and preventing its ubiquitination [232]. However, upon injury, Pin1 directly binds the phosphorylated Ser178-Pro motif in the death domain-associated protein and promotes its rapid degradation by the UPS [97]. This activates the ASK1/JNK pathway, specifically JNK3 [233,234], which facilitates the phosphorylation of Mcl-1 at Ser121-Pro and a subsequent conformational change that leads to Pin1 dissociation [232]. The consequent Mcl-1 degradation inhibits the B-cell lymphoma 2 protein, enabling the initiation of mitochondrial-mediated apoptosis via cytochrome c release [33,232]. Conversely, Pin1 upregulates neuronal apoptosis by stabilizing BIM_EL_ via binding at phosphorylated Ser65-Pro; uniquely, this pathway suppresses cell death in non-neural cells but increases apoptosis in neurons, likely due to the presence of the neuron-specific JNK scaffold protein JIP, which promotes Pin1 binding to BIM_EL_ [33].

These examples highlight the involvement of Pin1 in various neurological disorders, especially as Pin1 regulates various pro-apoptotic proteins, as described above, that are involved in neuronal death. Paradoxically, Pin1-mediated ubiquitination seems to play context-dependent, contrasting roles that are still not fully understood—for instance, promoting cell death in stroke, but binding protectively in spinal cord injury. Delineating the specific protein networks governed by Pin1 could provide invaluable roadmaps to understand disease triggers and guide therapeutic development. Thus, Pin1-mediated ubiquitination provides an intriguing perspective to developing therapeutics for a wide range of neurological dysfunctions.

## 5. Conformation-Specific Antibodies Target Intracellular Proteins for TRIM21-Mediated and Ubiquitination-Mediated Proteolysis

The neutralization ability of antibodies was long thought to be limited to the extracellular space, as they could not penetrate the cell membrane. With few exceptions, natural antibodies could not be used in live cells without permeabilization, making a wide range of intracellular targets beyond the reach of antibody-mediated detection, visualization, and inhibition [235]. Currently, most therapeutic antibodies are mostly restricted to extracellular membrane proteins or secreted proteins, which are estimated to make up 8000–9000 proteins [236] of the estimated 20,000 proteins in the human proteome [237]. This implies that targeting the intracellular proteome could unlock around two-thirds of the human proteome available for therapeutic intervention [236]. Among these intracellular targets are notorious cancer-driving intracellular targets, such as NF-kB and c-Myc, that have insofar been undruggable by small molecular inhibitors [238,239]. Although many antibodies cannot naturally penetrate live cells and require further engineering to acquire such a property [240], it has been shown that a subset of human and mouse antibodies produced in autoimmune disease patients [241,242,243,244,245] and in labs [246,247,248] can penetrate live cells and capture their intracellular antigens without engineering [234,249,250]. Monoclonal antibodies against many intracellular proteins, including oncoproteins [251,252], tau [253,254,255,256,257,258,259], TDP43 [260], and RANs [261], to name a few, are used as therapeutics without engineering, with some being evaluated in human trials [262]. Thus, intracellular antibodies represent a burgeoning extension of antibody therapeutics and research tools, allowing the study and treatment of many intracellular disease-driving targets.

Moreover, the recent discovery of TRIM21 introduces a mechanism by which intracellular antibodies may mediate their effects. TRIM21 is a cytosolic E3 Ub ligase and antibody receptor that acts as a final line of defense against invading viruses. Its physiological role is to intercept and neutralize antibody-coated viruses that have evaded extracellular immune defenses and entered the cell [263]. Specifically, TRIM21 recognizes the Fc of antibodies bound to invading pathogens and catalyzes K63-ubiquitin chain formation, stimulating proinflammatory pathways and anti-viral cell activity [264]. These pathogens are rapidly targeted for degradation by the proteasome and trigger innate immune activity, allowing TRIM21 to act as a bridge between adaptive and innate immune mechanisms [265].

In the context of Pin1 substrates, natural monoclonal antibodies have been developed that are able to reliably target cis or trans protein isomers inside the cell [266,267,268,269,270,271]. By targeting only one specific conformation of a protein for TRIM21-mediated protein degradation, it is possible to selectively deplete the toxic disease-driving isomers while retaining the beneficial forms (Figure 2). If used in combination with Pin1 inhibitors like Sulfopin [170] and a combination of arsenic trioxide and all-trans retinoic acid [169], which both block the further Pin1 generation of the target isomer, intracellular antibodies represent a fascinating prospect for treating certain Pin1-related pathologies.

Consistent the findings that Pin1 protects against tau-associated neurodegenerative diseases by catalyzing cis–trans isomerization, the accumulation of cis-P-tau has been shown to be an early biomarker and etiological factor for the neurodegenerative effects of traumatic and vascular brain injury and long-term sequelae like AD, chronic traumatic encephalopathy, and vascular dementia [266,267,268,269,270,271]. After severe or repetitive TBI in mice, cis P-tau mAb treatment not only eliminates cis P-tau231 and cistauosis but also prevents short- and long-term neuropathology and brain dysfunction [267,268,269]. In htau mice of AD, eliminating cis P-tau231 at a young age with cis mAb treatment prevented age-dependent tauopathy, neuronal loss, and learning and memory impairment [270]. Moreover, even after an NFT-like pathology and cognitive loss had already developed in aged htau mice, treating the mice with cis mAb still prevented cis P-tau231 accumulation and rescued the neuronal loss and brain atrophy [270]. Humanized cis P-tau mAb also reduced tau seeding and attenuated neuropathological and behavioral endpoints in transgenic mice expressing inducible mutant P301L tau, which has been linked with familial frontotemporal dementia [271]. In stroke models, eliminating cis P-tau231 using cis mAb rescues most stroke-like pathologies and brain dysfunction, and the efficacy is still strikingly obvious even after reducing the cerebral blood flow by ~50% for 6 months, which was confirmed by the unbiased single-cell transcriptomic profiling of VCID mice [270]. Thus, cis P-tau mAb might provide a prophylactic therapy that blocks the onset of neurodegeneration or a therapeutic intervention that is given after disease onset to reduce its severity in AD, TBI/CTE, and VCID [270]. A phase 1 trial of the i.v.-administered humanized cis P-tau231 mAb (PNT001) did not show obvious side effects and also produced CSF concentrations of PNT001 that suggest a potentially therapeutic effect [272].

Notably, TRIM21 mediates the neutralization of antibody-bound tau by promoting tau y ubiquitination and proteolysis via the proteasome and VCP unfoldase [268,273]. In mouse models, the antibody targeting of extracellular tau aggregates relied on TRIM21-mediated degradation after the internalization of the tau–antibody complex [274]. Antibody protection was lost in TRIM21-deficient mice [274].

Meanwhile, in immunotherapy for other diseases, antibodies have found massive success in pivotal advances such as immune checkpoint inhibitors and antibody–drug conjugates, yet their targets are still mostly relegated to the extracellular domain. The advent of stereospecific antibody technology that can capture intracellular targets and target them for TRIM21-mediated degradation introduces a huge potential for the treatment of a wide range of diseases in which dysfunctional Pin1 leads to the accumulation of a pathological protein. Because an abundance of Pin1 substrates, their regulatory mechanisms, and their pathological interactions have already been identified, future work will likely center around selecting appropriate molecular targets and developing stereospecific antibodies for translational experiments.

## Figures and Tables

**Figure 1 cells-13-00731-f001:**
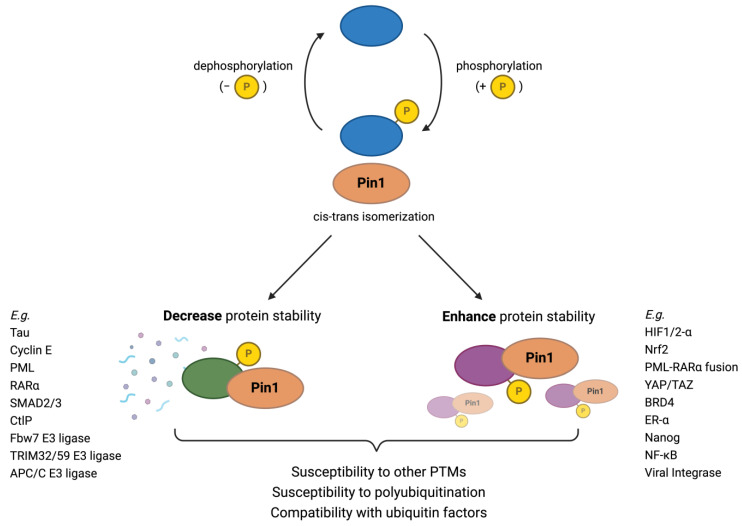
Overview of Pin1 involvement in ubiquitin-mediated degradation. After Pin1 binds and catalyzes a conformational change to a phosphorylated substrate, the resulting protein may have either decreased or increased susceptibility to ubiquitin-mediated degradation, depending on the molecular characteristics of the new conformation. Specifically, the resulting configuration may have an altered susceptibility to other post-translational mechanisms and polyubiquitination, as well as an affinity with ubiquitin factors. Furthermore, because Pin1 negatively regulates many E3 ligases, it may indirectly play a role in the stability of the substrate of these ubiquitin factors as well.

**Figure 2 cells-13-00731-f002:**
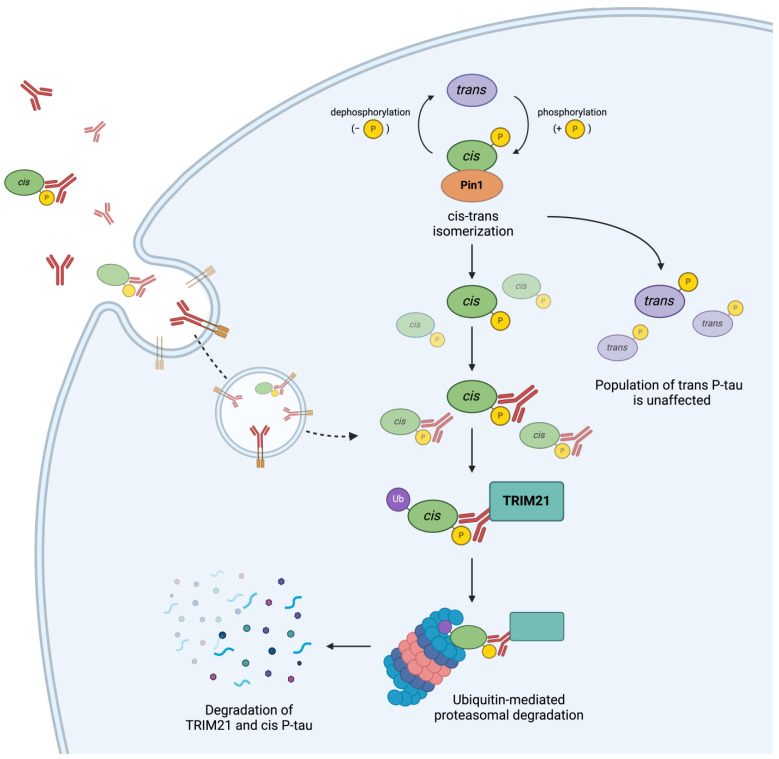
Stereospecific antibodies selectively deplete extracellular and intracellular cis P-tau. Pin1 plays a physiological role in safeguarding neurons against pathogenic cis-P-tau accumulation and neurofibrillary tangle formation by catalyzing the cis–trans isomerization of tau. However, when Pin1 is depleted during brain injury, cis-P-tau can accumulate and result in tau-associated neurodegeneration. Upon treatment with stereospecific cis-P-tau antibodies, cis-P-tau antibodies may bind to the p-Ser/Thr residues of their antigen and capture it extracellularly or intracellularly. Regardless, the antibody or antibody–antigen complexes enter the cell via Fc-receptor endocytosis. Finally, the TRIM21 E3 ligase engages the Fc region of the antibody, facilitating the ubiquitination and subsequent degradation of the TRIM21–antibody–antigen complex while sparing trans-P-tau or unphosphorylated, unaltered tau to carry out their physiological functions.

**Table 1 cells-13-00731-t001:** List of Pin1 substrates in which stability is altered upon Pin1 isomerization.

Substrate	Disease	References
**Increased Stability**		
β-catenin	Cancer	[22]
c-Myc	Cancer	[23,24,25,26]
Cyclin D1	Cancer	[27]
p53	Cancer	[28,29,30]
Bcl-2	Cancer	[31]
NF-kB	Inflammation	[10]
p73	Cancer	[32]
BIMEL	Huntington’s	[33]
Emi1	Cancer	[34]
HBx	HepB/Cancer	[35]
Her2	Cancer	[36,37]
Mcl-1	Cancer	[38]
Notch1	Cancer	[17,39]
AKT	Cancer	[40]
Cep55	Cancer	[41]
p27	Cancer	[42,43]
v-Rel	Cancer	[44]
NCID1	Cancer	[45]
Tax	HTLV/Cancer	[46,47]
COX-2	Inflammation	[48,49]
PPARγ	Cancer/Metabolism	[50]
Nanog	Cancer	[51]
Integrase Particle	HIV	[52]
4-Oct	Cancer	[53]
Mutant p53	Cancer	[54]
ADAR2	ALS	[55,56]
ERα	Cancer	[57,58]
HIPK2	Cancer	[59]
p63	Cancer	[60]
RBBP8	DNA Repair	[61]
Sp1	Cancer	[62]
HIF-1α	Cancer	[63,64,65]
ERα	Cancer	[66]
Separase	Cancer	[67]
PML-RARa	Cancer	[68]
CRMP2A	Alzheimer’s	[69]
PERIOD	Circadian Rhythm	[70]
BRD4	Cancer	[71]
p27	Cancer	[42]
Tissue Factor	Cancer	[72,73]
TRIM59	Cancer	[74]
YAP/TAZ	Cancer	[75,76]
ACC1	Cancer	[77]
BRCA1-BARD1	Cancer	[78]
FAAP20	DNA repair	[79]
BRCA1	Cancer	[80]
HBc	HepB/Cancer	[81]
HP1α	Cancer	[82]
Nrf2	Cancer	[83]
HIF-2α	Cancer	[84]
TRIM32	Cancer	[85]
HBV CP	HepB/Cancer	[81]
**Decreased Stability**		
Tau	Alzheimer’s	[86,87,88]
CF2	Cancer	[89]
RARa	Cancer	[90]
BTK	Cancer	[91]
IRF3	Cancer	[92]
Cyclin E	Cancer	[93,94,95,96]
Daxx	Cancer	[97]
SMRT	Cancer	[98]
FOXO4	Cancer	[99]
PML	Cancer	[100,101]
A3G	HIV	[102]
Smad2/3	Cancer	[103]
GRK2	Cancer	[104]
CDK10	Cancer	[105]
Fbw7	Cancer	[23,106]
PKC	Parkinson’s	[107]
Bora	Cancer	[108]
SUV39H1	Cancer	[109]
RUNX3	Cancer	[110]
KLF10	Cancer	[111]
Che-1	Cancer	[112]
REST	Neurodegeneration	[113]
Mutant HTT	Huntington’s	[114,115]
c-Jun	Cancer	[116]
IRF7	Inflammation	[117]
PRDM16	Metabolism	[20]
PSD-95	Epilepsy	[118]
STK3	Cancer	[75]
ATGL	Metabolism	[119]
SIK1	Cancer	[120]
pVHL	Cancer	[11]

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
