# Peer review of "Pin1-Catalyzed Conformation Changes Regulate Protein Ubiquitination and Degradation"

_cells, 2024, doi:10.3390/cells13090731_

Round 1
Reviewer 1 Report
Comments and Suggestions for Authors
Pin1 plays a crucial role in phosphorylation-mediated ubiquitination by regulating the stability and turnover of proteins. ​It catalyzes conformational changes in phosphorylated Ser/Thr-Pro motifs, enhancing protein stability by promoting exposure to E3 ubiquitin ligases or preventing their binding. Pin1 also regulates the length of the ubiquitin chain and modulates other post-translational modifications that interact with ubiquitination. This crosstalk between phosphorylation and ubiquitination is essential for controlling protein stability and turnover in various cellular processes and diseases. Pin1's role in polyubiquitination is important for regulating various cellular processes. ​Also, it affects the stability of oncogenes by modulating the substrate's compatibility with E3 ligase, negatively regulating substrate E3 ligases, modifying the length of a protein's Ub chain, and influencing the conformation and phosphorylation status of oncogenes. ​
The manuscript specifically highlights the role of Pin1, a prolyl isomerase, in regulating protein ubiquitination and degradation in the context of cancer and neurodegenerative diseases. The authors also proposed here a novel therapeutic approach using the ubiquitin-proteasome system for targeted degradation of intracellular tau aggregation. Overall, the abstract is well-written with a clear scope. The breadth of the review is good, as readability is well enough to provide a large readership. All subsections are discussed briefly and the texts are supported with important references.
However, I have some important suggestions to offer to improve the current manuscript:
1) “Pin1 is a critical mediator in the crosstalk between ubiquitination and phosphorylation”. In this context, authors are suggested to make a list the proteins' ubiquitination and phosphorylation, where Pin1 acts as a mediator and the related physiological role, in a tabular form, would help summarize the topic.
2) Pin1 expression and activity regulation during cancer or neurodegeneration could be important aspects to discuss. Also, the authors should elaborate on the cellular localization and tissue-specific distribution of Pin1, which would improve the scientific depth of the topic.
3) Pin1-mediated conformational changes serve as a regulatory mechanism for protein ubiquitination and degradation, contributing to the maintenance of cellular homeostasis and the regulation of various cellular processes. However, specific details of how Pin1 influences protein ubiquitination and degradation would depend on the context of individual proteins and cellular pathways. The authors should discuss this aspect in detail in a separate subsection. Some critical thoughts and open questions on this topic are critically missing. Authors must add some views on this.
Author Response
Reviewer 1:
- "Pin1 is a critical mediator in the crosstalk between ubiquitination and phosphorylation" In this context, authors are suggested to make a list the proteins' ubiquitination and phosphorylation where Pin1 acts as a mediator and the related physiological role, in a tabular form, would help summarize the topic.
- Thank you for your suggestion! A table has been generated of relevant Pin1 substrates that have altered stability due to cis-trans See Table 1 at the end of the document, in the figures section.
- Pin1 expression and activity regulation during cancer or neurodegeneration could be important aspects to discuss. Also, the authors should elaborate on the cellular localization and tissue-specific distribution of Pin1, which would improve the scientific depth of the topic.
- The reviewer is correct, a brief discussion on Pin1 distribution/localization during physiological and disease states has been added.
- Specifically, in the 3rd paragraph of the intro we discuss the physiological distribution of Pin1:
- “Accordingly, as Pin1 is a critical and dynamic signaling regulator for many substrates in essential functions such as the cell cycle, immunity, and metabolism [13-15], it is widely distributed at both the cellular and tissue levels depending on its substrates [16, 17]. Thus, Pin1 can be found in both the nucleus and cytoplasm of many cell types, including stromal, parenchymal, and stem cells [18-21].”
- In the 1st paragraph under the heading “Pin1-Regulated Protein Ubiquitination in Cancer”, we discuss how Pin1 levels change during cancer:
- “Typically, Pin1 is expressed at low levels in most normal cells such as fibroblasts and epithelial cells, although it may be highly expressed in some cell subsets of certain tissues [42]. However, in most solid malignancies like breast, prostate, and lung cancer [42-44], Pin1 is found to be commonly overexpressed when compared to their physiological counterparts [42]. One study found that in 38 of 60 human cancer types tested, Pin1 was overexpressed in 10% of those cases, suggesting that Pin1 overexpression is a critical event for tumorigenesis and may serve as an amplifier of oncogenic signaling [42]. While the mechanisms behind Pin overexpression are not fully understood, it has been posited that this trend is a result of the breakdown of transcriptional and post-transcriptional mechanisms that usually keep Pin1 tightly regulated [45, 46]. For instance, the deregulation of E2F during breast cancer promotes Pin1 expression [45], while the downregulation of the inhibitory phosphorylation of Pin1 leads to the accumulation of active Pin1 [46].”
- While Pin1 distribution in the nervous system and in AD is already discussed, we expanded on how Pin1 levels are changed during AD:
- “PIN1 is reduced in AD brain by tangle sequestration (1, 2), down-regulation (3-5), phosphorylation at S71 by DAPK1 (6-8), oxidation at C113 (9, 10), or even degradation induced by the environmental pollutant cobalt (11).”
- Pin1-mediated conformational changes serve as a regulator mechanism for protein ubiquitination and degradation, contributing to the maintenance of cellular homeostasis and the regulation of various cellular processes. However, specific details of how Pin1 influences protein ubiquitination and degradation would depend on the context of individual proteins and cellular pathways. The authors should discuss this aspect in detail in a separate subsection. Some critical thoughts and open questions on this topic are critically missing. Authors must add some views on this.
- Thank you for pointing this out, a paragraph regarding Pin1 functionality in relation to specific substrates and pathways has been added at the end of the introduction:
- “Pin1 functionality is critically dependent on the nature of its substrates and cellular pathways. Pin1 can interconvert from cis to trans conformation or vice versa depending on the identity of the substrate and relevant structural differences, since Ser/Thr-Pro motifs may have different preferences for being in cis or trans configuration after phosphorylation depending on the substate [23]. The trans conformation of peptide bonds tends to be more energetically favorable and common due to reduced steric hinderance from adjacent amino acids [24, 25]. Consequently, Pin1 typically catalyzes cis to trans isomerization in its substrates, although trans to cis substrates also exist [26-28] For example, the cadherin CDH1 is one notable substrate where Pin1 trans-cis isomerization contributes to changes in stability [28].”
Reviewer 2 Report
Comments and Suggestions for Authors
I found the review very interesting, well organized and the item well described.
I have just some suggestions:
- eliminate highlighting in the last paragraphs
- Add some figures or Tables to help in the description of the text, for example to list the targets of Pin1, with implications in diseases and references.
- lighten the fact that you propose a novel therapeutic strategy with this review. I think it is not appropriate for a review to propose a new therapy. It is not a paper that reports experimental data and these have already been published in the cited references
Author Response
Reviewer 2:
- Eliminate highlighting in the last paragraphs
- Thank you for pointing out those typos, the format has been unified across the document.
- Add some figures or Tables to help in the description of the text, for example to list the targets of Pin1, with implications in diseases and references.
- Thank you for your suggestion! A table has been generated of relevant Pin1 substrates that have altered stability due to cis-trans See Table 1 at the end of the document, in the figures section.
- Lighten the fact that you propose a novel therapeutic strategy with this review. I think it is not appropriate for a review to propose a new therapy. It is not a paper that reports experimental data and these have already been published in the cited references
- While we appreciate this comment, we believe it is important to tie the review content to a potential therapeutic application, especially as we discuss Pin1 in the development and treatment of disease at length in this article. Moreover, we consider the discussion of conformation-specific antibodies a natural extension of our discussion on Pin1-mediated ubiquitination because cis P-tau antibody has been shown to be valuable for early diagnosis and treatment of dementia in Alzheimer’s disease or after brain injury or stroke.
- Accordingly, we have expanded on this topic in the 4th paragraph under the heading “Conformation-specific Antibodies Target Intracellular Proteins for TRIM21-mediated Ubiquitination-mediated Proteolysis”:
- “While Pin1 usually protects against tau-associated neurodegenerative diseases by catalyzing the cis to trans isomerization, accumulation of cis-P-tau has been shown to be an early biomarker and etiological factor for the neurodegenerative effects of traumatic and vascular brain injury and long-term sequelae like AD and chronic traumatic encephalopathy and vascular dementia [207]. After severe or repetitive TBI in mice, cis P-tau mAb treatment not only eliminates cis P-tau231 and cistauosis, but also prevents short- and long-term neuropathology and brain dysfunction [210-212]. In htau mice of AD, eliminating cis P-tau231 at a young age with cis mAb treatment prevents age-dependent tauopathy, neuronal loss, and learning and memory impairment [211]. Moreover, even after NFT-like pathology and cognitive loss have already developed in aged htau mice, treating the mice with cis mAb still prevents cis P-tau231 accumulation and rescues neuronal loss and brain atrophy [211]. Humanized cis P-tau mAb also reduces tau seeding and attenuates neuropathological and behavioral endpoints in transgenic mice expressing inducible mutant P301L tau that has been linked with familial frontotemporal dementia [212]. In stroke models, eliminating cis P-tau231 using cis mAb rescues most stroke-like pathologies and brain dysfunction and the efficacy is still strikingly obvious even after reducing cerebral blood flow by ~50% for 6 months, which have been confirmed by the unbiased single-cell transcriptomic profiling of VCID mice [211]. Thus, cis P-tau mAb might provide a prophylactic therapy that blocks the onset of neurodegeneration or a therapeutic intervention that is given after disease onset to reduce its severity in AD, TBI/CTE and VCID. A phase 1 trials of the i.v. administered humanized cis P-tau231 mAb (PNT001) have not shown obvious side effects and also produced CSF concentrations of PNT001 that suggest a potentially therapeutic effect [213].”